# Alzheimer’s Disease CSF Biomarker Profiles in Idiopathic Normal Pressure Hydrocephalus

**DOI:** 10.3390/jpm12060935

**Published:** 2022-06-06

**Authors:** Salvatore Mazzeo, Filippo Emiliani, Silvia Bagnoli, Sonia Padiglioni, Lorenzo Maria Del Re, Giulia Giacomucci, Juri Balestrini, Assunta Ingannato, Valentina Moschini, Carmen Morinelli, Giulia Galdo, Cristina Polito, Camilla Ferrari, Gastone Pansini, Alessandro Della Puppa, Sandro Sorbi, Benedetta Nacmias, Valentina Bessi

**Affiliations:** 1Department of Neuroscience, Psychology, Drug Research and Child Health, Careggi University Hospital, University of Florence, Largo Brambilla, 3, 50134 Florence, Italy; salvatore.mazzeo@unifi.it (S.M.); filippo.emiliani@unifi.it (F.E.); silvia.bagnoli@unifi.it (S.B.); giuliagiacomucci.md@gmail.com (G.G.); juri.balestrini@unifi.it (J.B.); assunta.ingannato@unifi.it (A.I.); giulia.galdo@unifi.it (G.G.); camilla.ferrari@unifi.it (C.F.); alessandro.dellapuppa@unifi.it (A.D.P.); sandro.sorbi@unifi.it (S.S.); benedetta.nacmias@unifi.it (B.N.); 2IRCCS Fondazione Don Carlo Gnocchi, 50143 Florence, Italy; cristina.polito@unifi.it; 3Regional Referral Centre for Relational Criticalities, 50139 Florence, Italy; sonia_padiglioni@libero.it; 4Research and Innovation Centre for Dementia-CRIDEM, Careggi University Hospital, 50134 Florence, Italy; 5Research Unit of Medicine Ageing, Department of Experimental and Clinical Medicine, University of Florence, 50121 Florence, Italy; lorenzomaria.delre@unifi.it; 6Dipartimento Neuromosucolo-Scheletrico e Degli Organi di Senso, Careggi University Hospital, 50134 Florence, Italy; valentina.moschini02@gmail.com (V.M.); carmen.morinelli@studenti.unipd.it (C.M.); 7Neurosurgery Unit, Careggi University Hospital, Largo Giovanni Alessandro Brambilla 3, 50134 Florence, Italy; gastonepansini@gmail.com

**Keywords:** Alzheimer’s Disease, idiopathic normal pressure hydrocephalus, cerebrospinal fluid, biomarkers, cognitive impairment

## Abstract

Patients with idiopathic normal pressure hydrocephalus (iNPH) frequently show pathologic CSF Aβ_42_ levels, comparable with Alzheimer’s Disease (AD). Nevertheless, the clinical meaning of these findings has not been fully explained. We aimed to assess the role of AD CSF biomarkers (Aβ_42_, Aβ_42_/Aβ_40_, p-tau, t-tau) in iNPH. To this purpose, we enrolled 44 patients diagnosed with iNPH and 101 with AD. All the patients underwent CSF sampling. We compared CSF biomarker levels in iNPH and AD: Aβ_42_ levels were not different between iNPH and AD, while Aβ_42_/Aβ_40_, p-tau, and t-tau were significantly different and showed excellent accuracy in distinguishing iNPH and AD. A multiple logistic regression analysis showed that Aβ_42_/Aβ_40_ was the variable that most contributed to differentiating the two groups. Furthermore, iNPH patients with positive Aβ_42_/Aβ_40_ had higher p-tau and t-tau than iNPH patients with negative Aβ_42_/Aβ_40_. Those iNPH patients who showed cognitive impairment had lower Aβ_42_/Aβ_40_ and higher p-tau than patients without cognitive impairment. We concluded that positive CSF Aβ_42_ with negative Aβ_42_/Aβ_40_, p-tau, and t-tau is a typical CSF profile of iNPH. On the contrary, positive Aβ_42_/Aβ_40_ in iNPH patients, especially when associated with positive p-tau, may lead to suspicion of a coexistent AD pathology.

## 1. Introduction

Idiopathic normal pressure hydrocephalus (iNPH) is the most common form of hydrocephalus in adults with an estimated prevalence of 5.9% in patients over 80 years [1]. It is characterized by a classical triad of symptoms (cognitive disturbances, balance/gait impairment, and urinary incontinence), in the presence of a communicating hydrocephalus and a normal opening pressure upon lumbar puncture [2]. It is crucial to diagnose this condition early because 70–80% of patients show clinical improvement following ventricular–peritoneal shunt insertion [3]. Nevertheless, iNPH diagnosis could be challenging because the typical clinical triad is neither sensitive (it is present in <60% of patients [4,5]), nor specific, as clinical features of iNPH are shared with other cognitive and movement disorders, such as Alzheimer’s Disease (AD), vascular dementia, Parkinson’s disease, and atypical parkinsonian syndromes [6,7,8,9]. Moreover, cerebrospinal fluid (CSF) concentration of Aβ_42_ (one of the AD core biomarkers, whose reduction reflects Aβ amyloid plaques deposition in the brain [10]) was widely shown to be reduced also in iNPH [11]. In iNPH, the reduction of Aβ_42_ seems not to be related to Aβ amyloid plaques deposition, but to downregulation of Aβ production due to periventricular hypometabolism [12] and to the impairment of glymphatic clearance mechanisms and CSF turnover [13,14].

However, concomitant AD neuropathologic changes have been frequently seen in brain biopsies or in postmortem neuropathological examinations of patients with clinical diagnoses of iNPH [15,16,17]. Furthermore, the presence of AD pathology in iNPH was found to be associated with neuropsychiatric symptoms and behavior changes [18,19,20,21], as well as response to ventricular-peritoneal shunt [22,23,24,25,26]. Due to this evidence, a CSF AD biomarker analysis may have diagnostic and prognostic value, influencing the management of iNPH patients. 

Nevertheless, CSF Aβ_42_ measurement is influenced by interindividual physiological differences in amyloid processing [27] and false negatives are common also in patients with AD, as well as false positives in patients with other neurogenerative diseases and healthy individuals [28]. Measurement of the Aβ_40_ and Aβ_42_/Aβ_40_ ratio have been proposed to overcome this limit [28]. Aβ_40_ is the most abundant Aβ peptide and is less likely to aggregate than Aβ_42_. In AD, the reduced levels of Aβ_42_ are associated with slightly increased or steady levels of Aβ_40_ [29]. Therefore, the Aβ_42_/Aβ_40_ ratio is lower in AD than in healthy controls and showed higher accuracy as compared to Aβ_42_ in distinguishing AD from other neurodegenerative diseases [28]. In contrast, in iNPH patients, the CSF levels of all the amyloid precursor protein (APP) fragments (Aβ_38_, Aβ_40_, Aβ_42_, sAPPα, and sAPPβ) are decreased compared to controls [11,12,30]. Hence, we speculated that while Aβ_42_ levels are reduced in the CSF of iNPH patients, Aβ_42_/Aβ_40_ may be normal and could be the key factor in interpreting results of CSF biomarkers in iNPH. In this study, we aimed to test our hypothesis by assessing the accuracy of each CSF biomarker in distinguishing iNPH and AD patients to define a typical biomarker profile of iNPH.

## 2. Materials and Methods

### 2.1. Participants and Clinical Assessment

We included 44 patients diagnosed with iNPH according to international guidelines [7] and 101 patients diagnosed with AD according to NIA-AA criteria [31]. All of the patients were consecutively referred to the Centre for Alzheimer’s Disease and Adult Cognitive Disorders and Neurology Uniti of Careggi Hospital in Florence for CSF collection between December 2018 and March 2022. 

All patients in the iNPH group had ventricular enlargement associated with a patent Sylvian aqueduct and the absence of a macroscopic obstruction of CSF flow, lack of cortical atrophy, presence of periventricular water content, and an increased callosal angle in the coronal plane [32]. Patients in this group were not suffering from any other neurological, psychiatric, or medical conditions that could potentially explain their presenting symptoms. 

We excluded AD patients with a history of head injury, other neurological and/or systemic diseases, major depression, and alcoholism or other substance abuse. 

All of the patients underwent a comprehensive familial and clinical history, general and neurological examination, extensive neuropsychological investigation, brain magnetic resonance imaging (MRI) or brain Computed Tomography (TC), and lumbar puncture for CSF collection. Patients diagnosed with iNPH underwent a CSF tap test, a procedure that improves the diagnostic accuracy of iNPH and could predict a favorable response to CSF shunt surgery [33].

The local ethics committee approved the protocol of the study. All participants gave written informed consent to participate in the study.

### 2.2. CSF Tap Test

The iNPH patients underwent baseline evaluation of gait/balance and cognitive function within 24 h before the lumbar puncture. The baseline evaluation included: Short Physical Performance Battery [34], Mini-Mental State Examination [MMSE] [35], Frontal Assessment Battery [36], Phonemic Fluency Test, and Trail-making Test [37]. The lumbar puncture was performed at 9.00 a.m. with removal of 30–50 mL of CSF. The gait/balance assessment (Short Physical Performance Battery) was repeated 6, 24, and 48 h after the CSF subtraction. The neuropsychological examination (Phonemic Fluency test and Trail Making Test) was repeated 6, 24, and 48 h after the CSF subtraction. 

### 2.3. CSF Biomarkers Analysis

The CSF samples collected by the lumbar puncture were immediately centrifuged and stored at −80 °C until performing the analysis. Aβ_42_, Aβ_42_/Aβ_40_ ratio, t-tau, and p-tau have been measured using a chemiluminescent enzyme immunoassay (CLEIA) analyzer LUMIPULSE G600 (Fujirebio, Tokyo, Japan). Cut-off values for CSF were determined by following Fujirebio guidelines (Diagnostic sensitivity and specificity using clinical diagnosis and follow-up golden standard, 19 November 2018 [38]): Aβ_42_ > 670 pg/mL, Aβ_42_/Aβ_40_ ratio > 0.062, t-tau < 400 pg/mL and p-tau < 60 pg/mL. Patients were rated as Aβ_42_^+^ or Aβ_42_^−^ and Aβ_42_/Aβ_40_^+^ and Aβ_42_/Aβ_40_^−^ if the Aβ_42_ and Aβ_42_/Aβ_40_ were lower or higher than the cut-off values, respectively. Patients were rated as T^+^ or T^−^ and N^+^ or N^−^ if the CSF p-tau and t-tau concentrations were higher or lower than cut-off values, respectively [39].

### 2.4. Statistical Analysis

Patient groups were characterized using means and standard deviations (SD). We tested for normality in the distribution of the data using the Kolmogorov–Smirnov test. Depending on the distribution of the data, we used *t*-tests or non-parametric Mann–Whitney *U* tests for between-groups comparisons and Pearson’s r or Spearman’s ρ for correlations. We used chi-square tests to compare categorical data. We calculated the size effect with Cohen’s *d* for normally distributed numeric measures, η^2^ for Mann–Whitney *U* Test, and Cramer’s *V* for categorical data. Receiver operating characteristic (ROC) analyses were performed to evaluate the ability of CSF biomarkers to distinguish between iNPH and AD. Youden’s method was used to detect the best cut-off value and accuracy, sensitivity, and specificity. We used binomial logistic regression to ascertain the contribution of each biomarker in distinguishing iNPH and AD. Bonferroni correction was applied to correct for multiple comparisons. All statistical analyses were performed with SPSS software v.25 (SPSS Inc., Chicago, IL, USA) and the computing environment R 4.0.2 (R Foundation for Statistical Computing, Vienna, Austria, 2013).

## 3. Results

### 3.1. Description of the Sample

At onset, all the patients in the iNPH group had balance/gait impairment, 33 (75.00%) had cognitive impairment, and 31 (70.45%) had urinary incontinence. Twenty-three patients (52.27%) had the complete triad, while three patients (6.82%) only had balance/gait impairment. Thirteen out of 44 iNPH patients (29.55%) experienced an improvement in their gait after the CSF tap test, while 31 patients (70.45%) did not. Figure 1 shows correlations among the demographic features and CSF biomarkers in the iNPH and AD groups (Figure 1). Both in AD and iNPH, Aβ_42_/Aβ_40_ correlated with Aβ_42_ and p-tau correlated with t-tau. In the iNPH group Aβ_42_/Aβ_40_ also correlated with p-tau. In AD patients, MMSE was significantly correlated with age.

### 3.2. Comparison between iNPH and AD Groups

The iNPH patients were older than the AD (*p* = 0.044, Cohen’s *d* = −0.001) and had a lower frequency of *APOE* ε4^+^ (χ^2^ = 5.74, *p* = 0.017), but these differences where not statistically significant when adjusted for multiple comparisons (accepted at *p* < 0.003). There were no differences in years of education (*p* = 0.519, Cohen’s *d* = −0.135). AD patients had lower mean MMSE scores than iNPH patients (24.25 [SD = 4.02] vs. 19.57 [SD = 4.85], *p* < 0.001, Cohen’s *d* = 1.05). There were no differences in Aβ_42_ concentration between iNPH and AD (624.89 [316.30.82] pg/mL vs. 563.49 [236.941] pg/mL, *p* = 0.119, *d* = 0.22), while the Aβ_42_/Aβ_40_ ratios were significantly higher in iNPH than in AD (0.09 [0.02] vs. 0.04 [0.02], *p* < 0.001, *d* = 2.21). Patients in the iNPH group also had lower p-tau (32.13 [17.85] pg/mL vs. 125.52 [63.09] pg/mL, *p* < 0.001, *d* = 2.01) and t-tau (242.66 [205.59] pg/mL vs. 768.79 [374.22] pg/mL, *p* < 0.001, *d* = 1.74) concentrations than patients in the AD group (Table 1, Figure 2). 

Thirty-one out of 44 iNPH patients and 79 out of 101 AD patients had positive Aβ_42_ (CSF Aβ_42_ concentrations below the cut-off value), with no difference in distribution between groups (70.45% [95% C.I = 56.97:83.94] vs. 78.22% [95% C.I. 70.17:86.27], χ^2^ = 1.01, *p* = 0.315, *V* = 0.083). On the opposite, proportions of a positive Aβ_42_/Aβ_40_ ratio, (13.64% [95% C.I. = 3.50:23.78] vs. 93.07% [95% C.I. = 88.12:98.02], χ^2^ = 90.35, *p* < 0.001, *V* = 0.79), p-tau (6.82% [95% C.I. = 0:14.27] vs. 88.12% [95% C.I. 81.81:94.43%], χ^2^ = 87.35, *p* < 0.001, *V* = 0.78) and t-tau (11.36 [95% C.I. = 1.99:20.74] vs. 84.16% [95% C.I. = 77.04:91.28], χ^2^ = 68.98, *p* < 0.001, *V* = 0.69) concentrations were significantly lower in the iNPH patients compared to the AD group (Table 1, Figure 3).

### 3.3. CSF Biomarkers Accuracy in Distinguishing between iNPH and AD

Table 2 summarizes the area under the curve (AUC), accuracy, sensitivity, and specificity of each CSF biomarker in distinguishing iNPH and AD patients. We identified the cut-off values by Youden’s method. The Aβ_42_/Aβ_40_ ratio, p-tau, and t-tau showed very high accuracy without differences between the three biomarkers, as showed by the intersection of the 95% confidence intervals. Aβ_42_ was not able to distinguish between iNPH and AD (Figure 4).

### 3.4. Logistic Regression Models

To ascertain the effect of each biomarker in discriminating between iNPH and AD adjusting for age, we performed a multivariate logistic regression model including age and CSF biomarker concentrations as independent variables. The regression model was statistically significant (χ^2^ = 137.19, *p* < 0.001). The model explained 86.53% (Nagelkerke R^2^) of the variance in progression. The accuracy of the model in distinguishing between AD and iNPH was 93.79% (95% C.I. = 89.86:97.72) (sensitivity = 96.04% [95% C.I. = 92.87:99.21], specificity = 88.64% [95% C.I. = 83.47:93.81]). The Aβ_42_/Aβ_40_ ratio was shown as the only variable which significantly contributed to the model, independent of confounding factors (B = −128.38, S.E. = 47.83, *p* = 0.007) (Table 3).

We performed the same analysis considering CSF biomarkers after dichotomization according to cut-off values (Table 3). The regression model was statistically significant (χ^2^ = 128.67, *p* < 0.001). The model explained 83.21% (Nagelkerke R^2^) of the variance in progression. Age (B = −0.15, S.E. = 0.06, *p* = 0.013), Aβ_42_/Aβ_40_ (B = 4.37, S.E. = 1.06, *p* < 0.001) and p-tau (B = 5.05, S.E. = 1.58, *p* = 0.001) significantly contributed to the model (Table 3).

### 3.5. Comparison between iNPH and AD with Positive Aβ_42_

To explore the meaning of positive Aβ_42_ in iNPH patients, we compared the AD and iNPH groups and considered only patients who had positive Aβ_42_ (31 iNPH/Aβ_42_^+^ vs. 79 AD/Aβ_42_^+^, Table 4). The iNPH/Aβ_42_^+^ had higher MMSE than AD/Aβ_42_^+^ (24.68 [3.43] vs. 19.53 [3.42], *p* < 0.001, d = 1.50). There was no difference in age. The lower frequency of *APOE* ε4 in iNPH than in AD/Aβ_42_^+^ patients was not statistically significant when adjusted for multiple comparison (23.81 [95% C.I. = 5.59:42.03], χ^2^ = 4.52, *p* = 0.033, V = 0.22). The CSF biomarker concentrations (Aβ_42_/Aβ_40_, p-tau, and t-tau) were different between iNPH/Aβ_42_^+^ and AD/Aβ_42_^+^ (Figure 5). In particular among 31 iNPH/Aβ_42_^+,^ only six patients (19.35% [95% C.I. = 5.45:33.26]) had positive Aβ_42_/Aβ_40,_ two patients (6.45% [95% C.I. = 0:15.10]) had positive p-tau, and four patients (12.90% [95% C.I. = 1.10:24.70]) had positive t-tau. In contrast, 76 out of 79 AD/Aβ_42_^+^ patients had positive Aβ_42_/Aβ_40_ (96.20% [95% C.I. = 91.99:100]), 69 had positive p-tau (87.34% [95% C.I. = 80.01:94.67]), and 66 had positive t-tau (83.54% [95% C.I. = 75.37:91.72]). 

### 3.6. Features of iNPH Patients Classified According to Aβ_42_ and Aβ_42_/Aβ_40_ Status

We classified iNPH patients according to Aβ_42_ status (30 Aβ_42_^+^ and 14 Aβ_42_^−^). We found no differences in the demographic features and the Aβ_42_/Aβ_40_, p-tau, and t-tau values. *APOE* ε4 allele was not associated with positive Aβ_42_ status. There were no differences in proportion of cognitive impairment and urinary incontinence between Aβ_42_^+^ and Aβ_42_^−^. When we compared patients according to Aβ_42_/Aβ_40_ status (6 Aβ_42_/Aβ_40_^+^ and 38 Aβ_42_/Aβ_40_^−^, Table 5), patients with positive Aβ_42_/Aβ_40_ had higher p-tau (59.46 pg/mL [23.74] vs. 27.47 [11.81], *p* < 0.001, *d* = 1.87) than Aβ_42_/Aβ_40_^−^ patients. Furthermore, Aβ_42_/Aβ_40_^+^ had higher frequencies of positive p-tau (33.33% [95% C.I. = 0:71.05] vs. 2.63% [C.I. 95% = 0:7.72, χ^2^ = 7.68, *p* = 0.006, *V* = 0.41) and t-tau (50.00% [95% C.I. = 9.99:90.01] vs. 5.26% [C.I. 95% = 0:12.36, χ^2^ = 10.29, *p* = 0.001, V = 0.48) as compared to Aβ_42_/Aβ_40_^−^. Notably, all the patients in the Aβ_42_/Aβ_40_^+^ group had positive Aβ_42_, while four had negative p-tau. Only one patient with positive p-tau had negative Aβ_42_/Aβ_40_. Urinary incontinence was more frequent in the Aβ_42_/Aβ_40_^−^ group (76.32% [95% C.I. = 62.80:89.83] vs. 33.33% [95% C.I. 4.39:71.05], χ^2^ = 4.56, *p* = 0.032, *V* = 0.323), while there were no differences in cognitive impairment between Aβ_42_/Aβ_40_^+^ and Aβ_42_/Aβ_40_^−^. Particularly in the Aβ_42_/Aβ_40_^+^ group, all the patients had cognitive impairment, but only two out of the six had urinary incontinence.

### 3.7. Association between Clinical Features and CSF Biomarkers

The iNPH patients who showed cognitive impairment had lower Aβ_42_/Aβ_40_ (0.087 [0.022] vs. 0.097 [0.009], *p* = 0.036, *d* = 0.61) and higher p-tau (35.48 [18.00] vs. 22.08 [13.62], *p* = 0.029, *d* = 0.84) than patients without cognitive impairment at onset. CSF biomarker concentrations were not associated with urinary incontinence, nor with response to CSF tap test.

## 4. Discussion

Our first result confirmed that patients with iNPH have CSF Aβ_42_ concentrations comparable to AD and lower concentrations of p-tau and t-tau than AD, as widely described in previous studies [21,40,41,42]. In more detail, about 70% of patients in our series had positive CSF Aβ_42_, while p-tau and t-tau were positive in 7% and 11% of patients, respectively, which is consistent with a previous report by Santangelo et al. [21]. In contrast, we showed that Aβ_42_/Aβ_40_ was higher in iNPH than AD. Only 14% of iNPH patients had a Aβ_42_/Aβ_40_ ratio in the pathologic range (below the cut-off value), compared to 93% in the AD group.

When we estimated the diagnostic values of each biomarker, Aβ_42_/Aβ_40_, p-tau, and t-tau had very good accuracy in distinguishing between iNPH and AD, with no significant differences between the biomarkers. The logistic regression analysis demonstrated that Aβ_42_/Aβ_40_ and p-tau contributed to distinguishing iNPH patients from AD patients, in line with other works [43,44], with the notion that these biomarkers are more specific for AD than Aβ_42_ and t-tau [45,46]. Nevertheless, we would like to point out that, among iNPH patients with positive Aβ_42_/Aβ_40_, four had negative p-tau, while only one iNPH with positive p-tau had negative Aβ_42_/Aβ_40_. For the purposes of clinical practice, this remark mainly addresses to consider Aβ_42_/Aβ_40_ to identify AD pathology in iNPH patients and to use p-tau as a support biomarker. Based on the evidence that p-tau and t-tau can be increased in iNPH in association with long disease duration [47] and poor response at the CSF tap test [48], other authors suggested that p-tau and t-tau biomarkers may be prognostic factors more than diagnostic tools [49]. We aim to test this hypothesis and clarify the role of p-tau and t-tau in further works with wider samples. Notably, the cut-off values identified in our sample by automatized method were consistent with the cut-off values adopted according to LUMIPULSE producer guidelines. This result suggests using the same cut-off values adopted in clinical practice for the diagnosis of AD, as well as to distinguish iNPH from AD. 

Even though several studies already showed that Aβ_42_ is lower in iNPH patients than in healthy controls, the discrepancy between Aβ_42_ and Aβ_42_/Aβ_40_ was shown only by a few previous studies [43,50]. Our results confirm these reports and are in line with the evidence that in iNPH patients, the CSF levels of all the amyloid precursor protein (APP) fragments (Aβ_38_, Aβ_40_, Aβ_42_, sAPPα, and sAPPβ) are decreased compared to controls [11,12,30]. In contrast, the levels of Aβ42 in AD are associated with slightly increased or steady levels of Aβ40 [29]. Consequently, the ratio between Aβ_42_ and Aβ_40_ is supposed to be normal in iNPH patients without AD copathology, as reported by previous studies [43,50] and described in our sample. In particular, we showed that in 86% of iNPH patients with pathologic Aβ_42_, the Aβ_42_/Aβ_40_ ratio was normal, suggesting that in the majority of cases, the reduction of Aβ_42_ is associated with an equal reduction of Aβ_40_. Furthermore, we showed that iNPH patients with positive Aβ_42_ had lower CSF p-tau and t-tau concentrations than AD patients. In particular, only four out of 31 iNPH patients with positive Aβ_42_ also had positive t-tau and two out of these patients had positive p-tau. This evidence may support the hypothesis that Aβ_42_ in iNPH patients does not indicate AD copathology. 

The role of positive Aβ_42_/Aβ_40_ in iNPH is more difficult to interpret. We found that iNPH patients with positive Aβ_42_/Aβ_40_ had higher p-tau and t-tau concentrations than iNPH patients with negative Aβ_42_/Aβ_40_. In line with a previous result [43], we also found that iNPH patients who showed cognitive impairment had lower Aβ_42_/Aβ_40_ and higher p-tau than patients without cognitive impairment. Moreover, 100% of iNPH patients with positive Aβ_42_/Aβ_40_ also had positive Aβ_42,_ which might suggest that, in this group of patients, the low Aβ_42_/Aβ_40_ ratio is associated with a higher reduction of Aβ_42_ than Aβ_40_, as found in the AD patients [29]. Finally, we found that urinary incontinence was uncommon in iNPH with positive Aβ_42_/Aβ_40_ (only two out of six), as most of the patients in this group showed only gait/balance disturbance and cognitive impairment, suggesting a possible misdiagnosis. Considering this evidence, positive Aβ_42_/Aβ_40_ might indicate a coexistent AD pathology in patients affected by iNPH or a diagnosis of AD dementia. 

This hypothesis is supported by many studies which demonstrated that the Aβ_42_/Aβ_40_ ratio is more accurate than Aβ_42_ in identifying AD pathology [51,52].

If confirmed by further works, our results might suggest consideration of positive Aβ_42_ with negative Aβ_42_/Aβ_40_ and p-tau as the typical CSF profile of iNPH. On the contrary, positive CSFAβ_42_/Aβ_40_ should lead to suspicion of an underlying AD pathology.

This study had some limitations: (i) the relatively small sample size, especially when we classified patients according to Aβ_42_/Aβ_40_ status, which limits the impact of our conclusions; (ii) quantitative scores of gait/balance assessment are not available; (iii) data about gait/balance disturbance and urinary incontinence are not available for the AD group. However, we provided several pieces of evidence which may serve as starting points for future studies. As already stated, this is one of the first studies assessing Aβ_42_/Aβ_40_ in differential diagnostics of iNPH. Moreover, despite the small sample size, this is the first study to have classified patients according to Aβ_42_/Aβ_40_ status. Many studies divided iNPH patients according to their Aβ status, but did not distinguish between Aβ_42_ and Aβ_42_/Aβ_40_. As shown, Aβ_42_ could not be considered as an index of Aβ pathology in iNPH. On the other hand, considering only Aβ_42_/Aβ_40_ allowed us to classify iNPH patients as carriers or non-carriers of Aβ pathology with a higher accuracy.

## 5. Conclusions

We showed that positive CSF Aβ_42_ with negative Aβ_42_/Aβ_40_, p-tau, and t-tau is a frequently encountered finding in iNPH and should be considered as a typical CSF profile of iNPH. On the other hand, positive Aβ_42_/Aβ_40_ is uncommon in iNPH and is associated with a prevalent cognitive syndrome. Our results are in line with previous evidence and suggest that clinicians should not diagnose AD pathology in patients with iNPH and isolated positive CSF Aβ_42_. On the contrary, positive Aβ_42_/Aβ_40_ in iNPH patients, especially when associated with positive p-tau, may lead to suspicion of a coexistent AD pathology or revision of the diagnosis of iNPH.

## Figures and Tables

**Figure 1 jpm-12-00935-f001:**
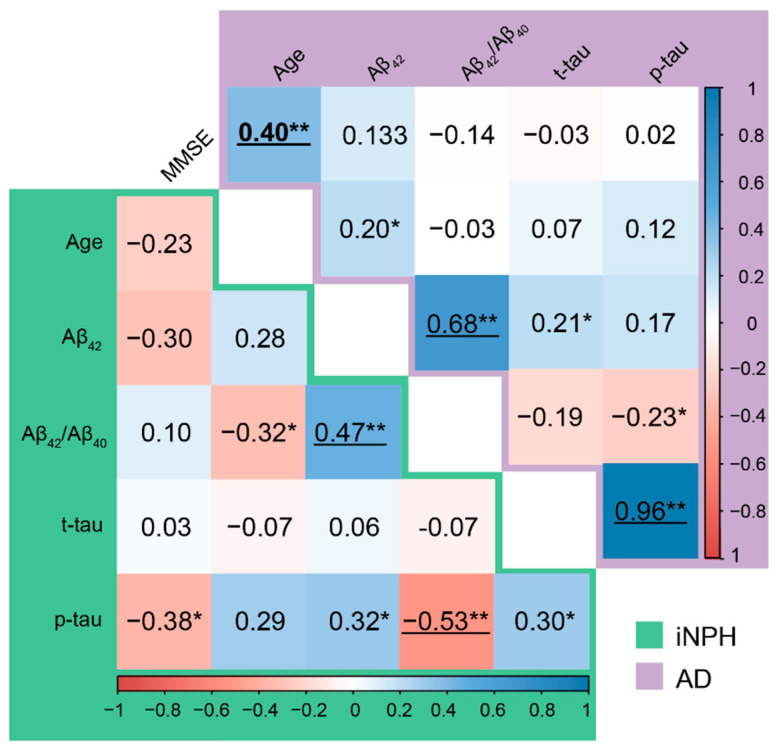
Correlation matrix. Values quoted in the correlation matrix are Pearson’s *r* correlation coefficients. Statistical significance received a Bonferroni adjustment and was accepted at *p* < 0.01 (significant correlations were reported as underlined characters). Color maps represent Pearson’s *r* correlation coefficients. * *p* < 0.05, ** *p* < 0.01.

**Figure 2 jpm-12-00935-f002:**
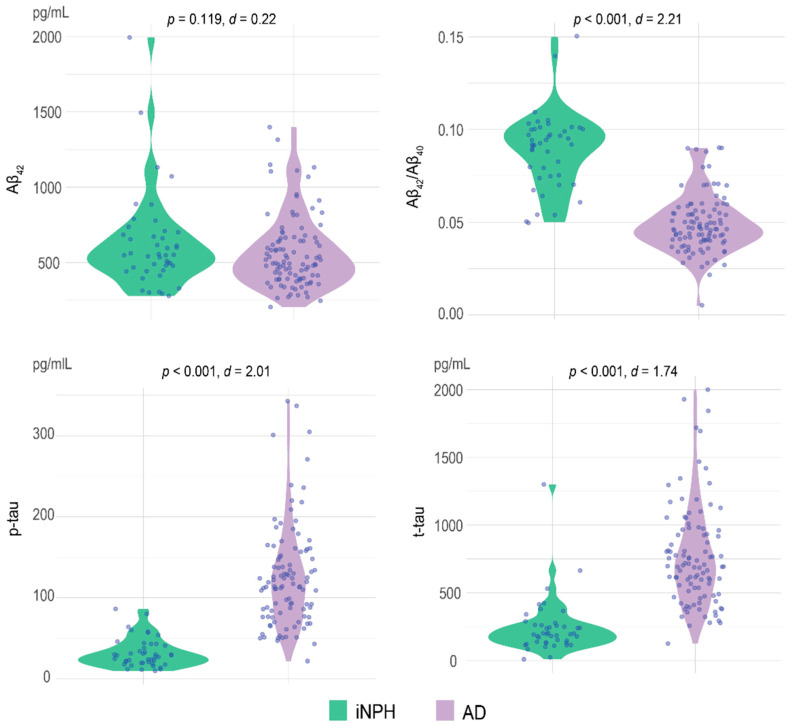
Comparison of the CSF biomarker concentrations and Aβ_42_/Aβ_40_ ratio between iNPH and AD. Values quoted on the *y*-axis indicate the CSF concentration (expressed as pg/mL) for Aβ_42;_ p-tau, t-tau, and the value of the ratio for the Aβ_42_/Aβ_40_. *p*-values and Cohen’s *d* are reported. Statistical significance received a Bonferroni adjustment and was accepted at *p* < 0.003.

**Figure 3 jpm-12-00935-f003:**
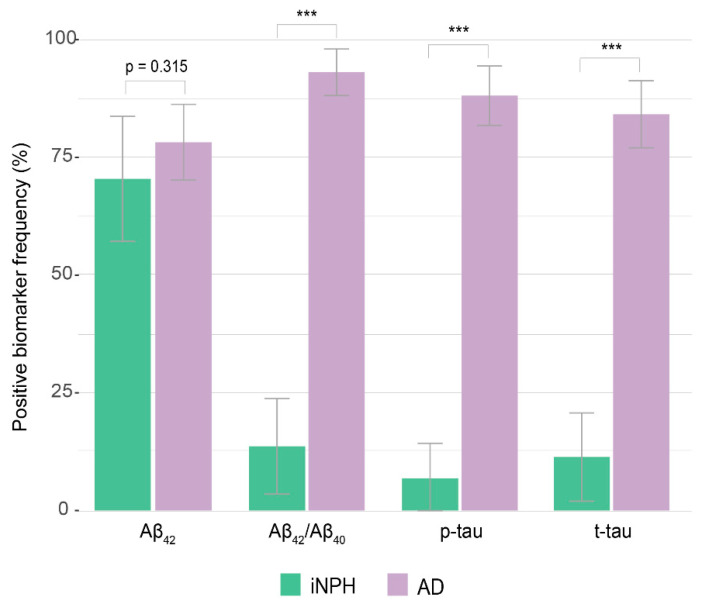
Relative frequencies of positive Aβ_42_, Aβ_42_/Aβ_40_, p-tau, and t-tau in iNPH and AD. Statistical significance received a Bonferroni adjustment and was accepted at *p* < 0.003. *** *p* < 0.001.

**Figure 4 jpm-12-00935-f004:**
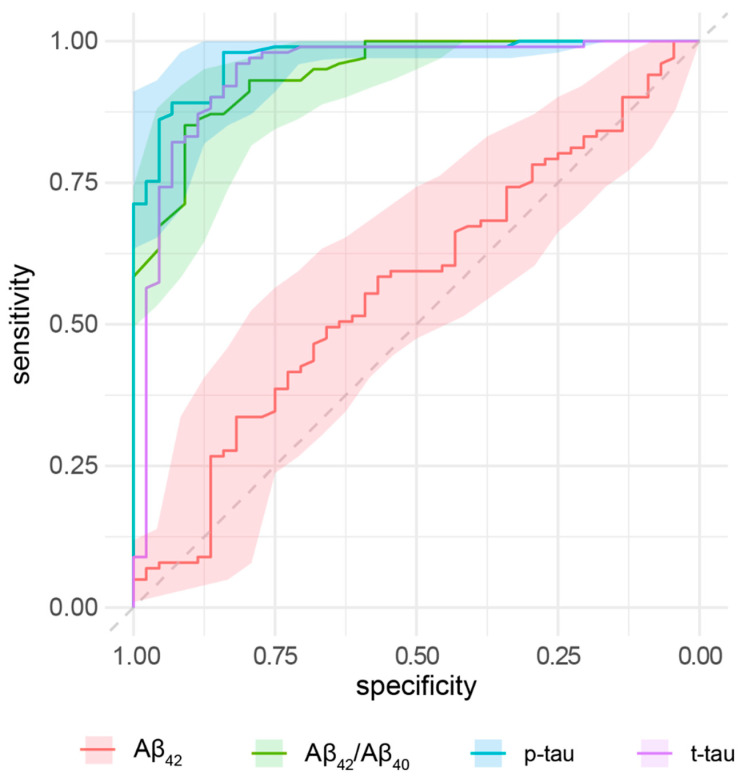
ROC curves for the accuracy of Aβ_42_, Aβ_42_/Aβ_40_, p-tau, and t-tau in distinguishing iNPH and AD. Colored shapes indicate 95% C.I.

**Figure 5 jpm-12-00935-f005:**
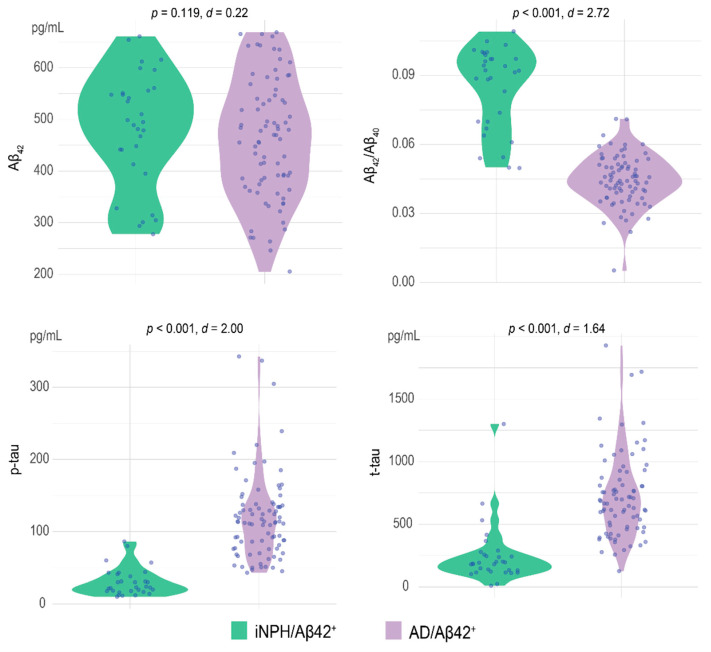
Comparison of CSF biomarker concentrations and Aβ_42_/Aβ_40_ ratio between iNPH/Aβ42^+^ and AD/Aβ42^+^. The values quoted on the *y*-axis indicate the CSF concentration (expressed as pg/mL) for Aβ_42_, p-tau, and t-tau.

**Table 1 jpm-12-00935-t001:** Comparison of demographic variables and CSF biomarker values between iNPH and AD patients.

	iNPH	AD
N	44	101
Age, mean (SD)	73.92 (7.42)	71.28 (6.83)
Sex (women/men)	26/18	50/51
Years of education, mean (SD)	9.71 (3.94)	9.98 (4.21)
*APOE* ε4^+^, % (95 CI %)	24.14 (8.56:39.71)	49.45 (39.18:59.72)
MMSE, mean (SD)	24.25 (4.02) ^a^	19.57 (4.85) ^a^
Aβ_42_ (pg/mL), mean (SD)	624.89 (316.30)	563.49 (236.94)
Aβ_42_/Aβ_40_, mean (SD)	0.09 (0.02) ^b^	0.04 (0.02) ^b^
p-tau (pg/mL), mean (SD)	32.13 (17.85) ^c^	125.52 (63.09) ^c^
t-tau (pg/mL), mean (SD)	242.66 (205.57) ^d^	768.79 (374.22) ^d^
Aβ_42_^+^, % (95% C.I.)	70.45 (56.97:83.94)	78.22 (70.17:86.27)
Aβ_42_/Aβ_40_^+^, % (95% C.I.)	13.64 (3.50:23.78) ^e^	93.07 (88.12:98.02) ^e^
T^+^, % (95% C.I.)	6.82 (0:14.27) ^f^	88.12 (81.81:94.43) ^f^
N^+^, % (95% C.I.)	11.36 (1.99:20.74) ^g^	84.16 (77.04:91.28) ^g^

Values quoted in the table are mean (±SD) or *n* (%). Statistical significance received a Bonferroni adjustment and was accepted at *p* < 0.003). ^a^
*p* < 0.001, *d* = 1.05; ^b^
*p* < 0.001, *d* = 2.21; ^c^
*p* < 0.001, *d* = 2.01; ^d^
*p* < 0.001, *d* = 1.74; ^e^ χ^2^ = 90.35, *p* < 0.001, *V* = 0.79; ^f^ χ^2^ = 87.35, *p* < 0.001, *V* = 0.78; ^g^ χ^2^ = 68.98, *p* < 0.001, *V* = 0.69.

**Table 2 jpm-12-00935-t002:** CSF biomarker accuracy.

	Cut-Off	AUC	Accuracy, % (CI.95%)	Sensitivity, % (CI.95%)	Specificity, % (CI.95%)
Aβ_42_	776.34	0.567	64.14 (56.33:71.95)	84.16 (78.22:90.10)	18.18 (11.90:24.46)
Aβ_42_/Aβ_40_	0.068	0.943	86.21 (80.60:91.82)	87.13 (81.68:92.58)	84.09 (78.14:90.04)
p-tau	65.24	0.969	88.28 (83.04:93.52)	95.45 (92.06:98.84)	85.15 (79.36:90.94)
t-tau	509.09	0.941	80.69 (74.27:87.11)	93.18 (89.08:97.28)	75.25 (68.23:82.27)

Cut-off values were estimated by Youden’s method. Area under the curve (AUC), accuracy, sensitivity, and specificity for each biomarker are reported. Accuracy, sensitivity, and specificity are expressed as percentages (95% C.I.).

**Table 3 jpm-12-00935-t003:** Multivariate logistic regression models.

	B	S.E.	*p*	OR	95% C.I.	
					Lower	Upper
CSF biomarkers (quantitative values)						
Age	−0.18	0.08	0.024	0.83	0.71	0.98
Aβ_42_	0.01	0.00	0.081	1.00	0.99	1.01
**Aβ_42_/Aβ_40_**	**−128.38**	**47.83**	**0.007**	**2.013 × 10^−31^**	**3.44 × 10^−97^**	**9.03 × 10^−8^**
p-tau	0.04	0.03	0.193	1.04	0.98	1.09
t-tau	0.00	0.00	0.347	1.07	.99	1.01
χ^2^ = 137.19, *p* < 0.001, Nagelkerke R^2^ = 86.53%						
CSF biomarkers (dichotomized values)						
Age	−0.15	0.06	0.013	0.86	0.76	0.97
A (Aβ_42_^+^)	−2.06	1.21	0.092	0.13	0.01	1.35
A (Aβ_42_/Aβ_40_^+^)	**4.37**	**1.06**	**<0.001**	**79.20**	**9.89**	**643.36**
**T^+^**	**5.05**	**1.58**	**0.001**	**155.78**	**7.09**	**3420.53**
N^+^	−1.28	1.46	0.381	0.278	0.02	4.86
χ^2^ = 128.67, *p* < 0.001, Nagelkerke R^2^ = 83.21%						

Regression Coefficients (B), Standard errors (S.E), *p*-value (*p*), Odds Ratio (OR), and 95% Confidence Intervals (95% C.I.) for covariates included in the logistic regression models are reported. Statistical significance received a Bonferroni adjustment and being accepted at the *p* < 0.01, highlighted in bold.

**Table 4 jpm-12-00935-t004:** Comparison of demographic variables and CSF biomarker values between iNPH/Aβ_42_^+^ and AD/Aβ_42_^+^ patients.

	iNPH/Aβ_42_^+^	AD/Aβ_42_^+^
N	31	79
Age, mean (SD)	73.08 (8.24)	70.95 (6.97)
Sex (women/men)	19/12	37/42
Years of education, mean (SD)	9.73 (3.77)	10.21 (4.39)
*APOE* ε4^+^, % (95 CI %)	23.81 (5.59:42.03)	50.00 (38.45:61.55)
MMSE, mean (SD)	24.68 (4.02) ^a^	19.53 (3.42) ^a^
Aβ_42_ (pg/mL), mean (SD)	482.48 (110.44)	462.76 (116.18)
Aβ_42_/Aβ_40_, mean (SD)	0.08 (0.12) ^b^	0.04 (0.01) ^b^
p-tau (pg/mL), mean (SD)	30.44 (19.10) ^c^	120.02 (60.25) ^c^
t-tau (pg/mL), mean (SD)	244.74 (238.07) ^d^	724.65 (338.935) ^d^
Aβ_42_/Aβ_40_^+^, % (95% C.I.)	13.64 (3.50:23.78) ^e^	93.07 (88.12:98.02) ^e^
T^+^, % (95% C.I.)	6.82 (0:14.27) ^f^	88.12 (81.81:94.43) ^f^
N^+^, % (95% C.I.)	11.36 (1.99:20.74) ^g^	84.16 (77.04:91.28) ^g^

Values quoted in the table are mean (±SD) or *n* (%). Statistical significance received a Bonferroni adjustment and was accepted at *p* < 0.003). ^a^
*p* < 0.001, d = 1.50; ^b^
*p* < 0.001, *d* = 2.72; ^c^
*p* < 0.001, *d* = 2.00; ^d^
*p* < 0.001, *d* = 1.64; ^e^ χ^2^ = 69.29, *p* < 0.001, *V* = 0.79; ^f^ χ^2^ = 63.66, *p* < 0.001, *V* = 0.76; ^g^ χ^2^ = 48.01, *p* < 0.001, *V* = 0.66.

**Table 5 jpm-12-00935-t005:** Comparison of demographic variables and CSF biomarkers values between iNPH Aβ_42_/Aβ_40_^−^ and iNPH Aβ_42_/Aβ_40_^+^ groups.

	iNPH Aβ_42_/Aβ_40_^−^	iNPH Aβ_42_/Aβ_40_^+^
N (%)	38 (86.36%)	6 (13.64%)
Age, mean (SD)	73.36 (7.21)	77.61 (8.61)
Sex (women/men)	24/14	2/4
Years of education, mean (SD)	9.54 (3.94)	10.60 (4.28)
*APOE* ε4^+^, % (95 CI %)	20.00 (4.32:35.68)	50.00 (1.00:99.00)
MMSE, mean (SD)	24.52 (4.14)	22.80 (3.35)
Aβ_42_ (pg/mL), mean (SD)	644.66 (335.17)	499.67 (85.63)
p-tau (pg/mL), mean (SD)	27.47 (11.82) ^a^	59.46 (23.74) ^a^
t-tau (pg/mL), mean (SD)	222.29 (201.57)	371.67 (199.11)
A (Aβ_42_^+^), % (95% C.I.)	65.79 (50.71:80.87)	100
T^+^, % (95% C.I.)	2.63 (0:7.71) ^b^	33.33 (0:71.05) ^b^
N^+^, % (95% C.I.)	5.26 (0:12.36) ^c^	50.00 (9.99:90.01) ^c^
Cognitive impairment, % (95% C.I.)	71.05 (56.63:85.47)	100
Urinary incontinence, % (95% C.I.)	76.32 (62.80:89.83) ^d^	33.33 (4.39:71.05) ^d^
Response to CSF tap test, % (95% C.I.)	28.95 (14.53:43.37)	33.33 (0:71.05)

Values quoted in the table are mean (±SD) or *n* (%). Statistical significance received a Bonferroni adjustment and being accepted at the *p* < 0.004). ^a^
*p* < 0.001, d = 1.87; ^b^ χ^2^ = 7.68, *p* = 0.006, *V* = 0.41; ^c^ χ^2^ = 10.29, *p* = 0.001, *V* = 0.48; ^d^ χ^2^ = 4.56, *p* = 0.032, *V* = 0.323.

## Data Availability

An anonymized data that support the findings of this study will be shared by request from any qualified investigator.

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
