# Peer review of "Alzheimer’s Disease CSF Biomarker Profiles in Idiopathic Normal Pressure Hydrocephalus"

_jpm, 2022, doi:10.3390/jpm12060935_

Round 1

Reviewer 1 Report

Major concerns:

The main criticism of this work is that the aim of this study and what it is trying to address is not clear. The conclusion helps to clarify the aim of the study but this should be clear from the outset.

N-numbers become low when you look into the subgroups – are these studies adequately powered to enable conclusions to be drawn?

At the start of the discussion it states that CSF Ab42 was lower than the cut-off values in a very high proportion of iNPH patients but that only 14% of iNPH patients have a positive Ab42/40 ratio. In the conclusion it then states that positive CSF Ab42 with negative Ab42/40, p-tau and t-tau is a frequently encountered finding in iNPH. But if a high proportion of iNPH patients have ab42 levels below the cut-off how can this be the case? This needs to be explained more clearly.

Specific comments to address:

Line 74: In AD, the reduced levels of Ab42 are associated with slightly increased or steady levels of Ab40. Therefore the Ab42:Ab40 ratio is decreased in AD patients…. Is this in comparison to iNPH? Please clarify this as it is not clear what is meant here.

Line 85 and 86: What is a consecutive patient?

Figure 1: why are some values underlined? There is also a significant correlation between Age and MMSE in the AD patients – this is not commented on.

Line 163: iNPH and AD patients did not differ in APOE e4 allele frequency – Looking at the data in table 1 I’m not sure that this is accurate as ApoE e4 allele frequency appears to be much higher in the AD group. As ApoE e4 does not appear to be a risk factor for iNPH would you not expect a higher frequency in the AD population?

Results, e.g. line 166: Please explain how the data is presented and what the values are that are shown in the text

Figure 2 is labelled as Figure 1

The text in line 179-181 suggests that not all of the patients had detectable measurements made. Please indicate n-numbers for the different measurements in Table 1 Figure 2 and Figure 3

Line 193: should it read ‘Aβ42 was not able to distinguish between…’?

Line 208: should be ‘contributed’

Line 277: ‘In contrast…’ would be a better phrase here

Line 277-279: So does this mean that where there is a positive Ab ratio in iNPH this is higher than in AD, but not many patients have this positive ratio?

Discussion: iNPH Ab42 levels are reduced – but in comparison to what? To AD? Please make such statements clearer.

Author Response

We thank the reviewer for his suggestions that allowed us to considerably improve our manuscript.

Major concerns:

The main criticism of this work is that the aim of this study and what it is trying to address is not clear. The conclusion helps to clarify the aim of the study but this should be clear from the outset.

We thank the reviewer for this suggestion. We add a sentence in the Introduction to better explain our aims (line 81-83)

N-numbers become low when you look into the subgroups – are these studies adequately powered to enable conclusions to be drawn?

We agree with the reviewer, as we already stated in the discussion. We further stressed this point at line 338.

At the start of the discussion it states that CSF Ab42 was lower than the cut-off values in a very high proportion of iNPH patients but that only 14% of iNPH patients have a positive Ab42/40 ratio. In the conclusion it then states that positive CSF Ab42 with negative Ab42/40, p-tau and t-tau is a frequently encountered finding in iNPH. But if a high proportion of iNPH patients have ab42 levels below the cut-off how can this be the case? This needs to be explained more clearly.

Thank you for this comment. We stated in the discussion that the reduction of Aβ42 reflects Aβ amyloid plaques deposition in the brain in AD. Therefore, a CSF Aβ42 concentration below the cut-off value is considered to be positive for Aβ pathology. Following your suggestion, to make this point clearer, we modified the sentence in the discussion (line 285-286)

Specific comments to address:

Line 74: In AD, the reduced levels of Ab42 are associated with slightly increased or steady levels of Ab40. Therefore the Ab42:Ab40 ratio is decreased in AD patients…. Is this in comparison to iNPH? Please clarify this as it is not clear what is meant here.

We better explained this sentence (75-76)

Line 85 and 86: What is a consecutive patient?

We modified this point to explain that patients were consecutively referred to our center between December 2018 and March 2022 (all the patients evaluated in this period were included) (87-89).

Figure 1: why are some values underlined? There is also a significant correlation between Age and MMSE in the AD patients – this is not commented on.

Underlined values indicated significant correlations (we better explained in the caption). We also reported the significant correlation between MMSE and Age in the "Results" section (line 156).

Line 163: iNPH and AD patients did not differ in APOE e4 allele frequency – Looking at the data in table 1 I’m not sure that this is accurate as ApoE e4 allele frequency appears to be much higher in the AD group. As ApoE e4 does not appear to be a risk factor for iNPH would you not expect a higher frequency in the AD population?

We agree with the reviewer. We included in the text p-values and effect size for age and APOE (line 167-170). We pointed out that the differences regarding these two variables were not significant as we applied a Bonferroni adjustment for multiple comparisons, as explained in the method section. We also think that the lack of significance might be due to the small sample size. Nevertheless, we explained in the methods that we applied a Bonferroni correction (line 167-168)

Results, e.g. line 166: Please explain how the data is presented and what the values are that are shown in the text

We clarified in the text that reported values are means, standard deviations (in braquetes) and Cohen's d (169-172)

Figure 2 is labelled as Figure 1

Thank you for this comment, we corrected the typo.

The text in line 179-181 suggests that not all of the patients had detectable measurements made. Please indicate n-numbers for the different measurements in Table 1 Figure 2 and Figure 3

We are afraid there is a misunderstanding. All the patients had detectable measurements made. At line 178-179 reported that 31 out of 44 iNPH and 79 out of 101 AD patients had positive Aβ42, as they had a Aβ42 below the cut-off value to define Aβ42 concentration as positive. We better explained this point in the text (185-186).

Line 193: should it read ‘Aβ42 was not able to distinguish between…’?

We corrected as suggested by the reviewer (line 200)

Line 208: should be ‘contributed’

We corrected as suggested by the reviewer (line 215)

Line 277: ‘In contrast…’ would be a better phrase here

We modified as suggested by the reviewer (line 284)

Line 277-279: So does this mean that where there is a positive Ab ratio in iNPH this is higher than in AD, but not many patients have this positive ratio?

As stated in methods, a higher Ab42/Ab40 concentration corresponds to a less positive value. Therefore, iNPH patients had higher Ab42/Ab40 concentration than AD patients. Only 14% of iNPH patients had  Ab42/Ab40 ratio in the pathologic range (below the cut-off value). We better clarify this point (line 282-283)

Discussion: iNPH Ab42 levels are reduced – but in comparison to what? To AD? Please make such statements clearer.

We modified as suggested by the reviewer (line 309-314)

Reviewer 2 Report

The article “Alzheimer’s disease CSF biomarker profiles in idiopathic normal pressure hydrocephalus” is an interesting work on correlation between iNPH and AD. However, mentioned below are some queries which could help clarify a few doubts:

  1. In figure 1, what do the high levels of Aβ42/Aβ40 in iNPH compared to AD indicate even when Aβ42 levels in both cohorts are similar?

  1. What is “Aβ42/Aβ40, median (IQR)”, which is higher in iNPH than AD?

  1. Can the authors explain the meaning they are trying to convey through line no. 304 “Consequently, Aβ42/Aβ40 ratio is not supposed to be reduced in iNPH patients….”? Has it ever been reported earlier in any study?

Author Response

We thank the reviewer for his appreciation and for his useful suggestions

1.In figure 1, what do the high levels of Aβ42/Aβ40 in iNPH compared to AD indicate even when Aβ42 levels in both cohorts are similar?

We thank the reviewer for this question. We described some hypotheses to explain this discrepancy in the "Discussion" section. In particular (line 309), we better explained that "the levels of Aβ42 in AD are associated with slightly increased or steady levels of Aβ40" while in iNPH "the reduction of Aβ42 is associated with an equal reduction of Aβ40". Consequently, in AD Aβ42/Aβ40 ratio is supposed to be reduced in AD will in iNPH the Aβ42/Aβ40 ratio is supposed to remain constant.

2. What is “Aβ42/Aβ40, median (IQR)”, which is higher in iNPH than AD?

 We thank the reviewer. We corrected this typo.

3. Can the authors explain the meaning they are trying to convey through line no. 304 “Consequently, Aβ42/Aβ40 ratio is not supposed to be reduced in iNPH patients….”? Has it ever been reported earlier in any study?

We thank the reviewer. We modified the text at line 311-312

Round 2

Reviewer 1 Report

Comments and concerns have been addressed appropriately